

# Nanoparticle-based antifungal therapies innovations mechanisms and future prospects

Qinglin Wu[1,*], Fulan Cen[1,*], Ying Xie[2], Xianjia Ning[3], Jinghua Wang[3], Zhenghao Lin[1] and Jia Huang[1]

[1] Department of Intensive Care Unit, Shenzhen Third People's Hospital and the Second Hospital Affiliated with the Southern University of Science and Technology, Shenzhen, Guangdong Province, China
[2] Graduate School of Public Administration, Seoul National University, Seoul, Republic of South Korea
[3] Center of Clinical Epidemiology, Shenzhen Third People's Hospital and the Second Hospital Affiliated with the Southern University of Science and Technology, Shenzhen, Guangdong Province, China
[*] These authors contributed equally to this work.

## ABSTRACT

Fungal infections present an increasing global health challenge, with a substantial annual mortality rate of 1.6 million deaths each year in certain situations. The emergence of antifungal resistance has further complicated treatment strategies, underscoring the urgent need for novel therapeutic approaches. This review explores recent advances in nanoparticle-based therapies targeting fungal infections, emphasizing their unique potential to enhance drug solubility, bioavailability, and targeted delivery. Nanoparticles offer the ability to penetrate biological barriers, improve drug stability, and act as direct antifungal agents by disrupting fungal cell walls and generating reactive oxygen species. Despite their promising applications, challenges such as potential toxicity, scalability of production, and the need for controlled drug release remain. Future research should focus on optimizing nanoparticle properties, evaluating long-term safety profiles, developing environmentally sustainable synthesis methods, and exploring synergistic approaches with existing antifungal drugs. Nanotechnology offers a transformative opportunity in the management of fungal diseases, paving the way for more effective and targeted treatments.

## INTRODUCTION

The field of nanoparticle-based antifungal medicines provides novel answers to the rising issue of fungal infections and drug resistance. Fungal infections are an increasing public health concern, resulting in substantial mortality worldwide. The World Health Organization estimates that fungal illnesses cause 1.6 million deaths each year, with invasive infections playing a significant role. Emerging antifungal resistance complicates therapy, emphasizing the need for novel methods (*Fisher et al., 2022*). Traditional antifungal drugs are frequently associated with concerns such as limited effectiveness, poor absorption, toxicity, and resistance. The extensive use of azoles has resulted in substantial resistance in *Candida* species, necessitating the search for other treatments (*Whaley et al., 2016*).

Corresponding authors
Zhenghao Lin, linzh2019@163.com
Jia Huang, hjsunnyvale@hotmail.com

Nanotechnology offers a number of benefits, including enhanced medicine distribution, solubility, and targeted action, as well as direct antifungal effects such fungal cell destruction and the formation of reactive oxygen species (ROS) (*Halbandge et al., 2019*). Despite these potential applications, there are several barriers to the clinical translation of nanoparticle-based therapies. These include concerns with safety, toxicity, manufacturing scalability, and controlled drug release (*Metselaar & Lammers, 2020*). This review is useful for researchers working in antifungal treatment and related topics such as medication delivery and infectious disorders. It reviews recent accomplishments, identifies obstacles, and offers future research areas that may help produce more effective and long-lasting antifungal treatments (*Dordevic et al., 2022*).

## SURVEY METHODOLOGY

**Search engines and databases:** We utilized reputable scientific databases, including PubMed, Web of Science, Scopus, and Google Scholar. These platforms provide extensive access to peer-reviewed articles, ensuring coverage of a wide range of studies across disciplines. **Search Terms:** We employed specific and comprehensive search terms to capture all relevant publications. These included: "Nanoparticles AND antifungal therapy", "Drug resistance AND fungal infections", "Nanotechnology AND fungal pathogens", "Metallic nanoparticles AND antifungal mechanisms", "Targeted drug delivery AND fungal infections", "Reactive oxygen species AND fungal cell walls", *etc*. Boolean operators and combinations of keywords ensured that we included all pertinent studies while avoiding irrelevant ones. **Inclusion Criteria:** Articles were selected based on the following: Focus on antifungal therapies, particularly those involving nanoparticles. Studies detailing the mechanisms of nanoparticle action, drug delivery systems, or resistance in fungal pathogens. Peer-reviewed journal articles, reviews, and meta-analyses published mainly in the last decade to ensure relevance. Publications presenting original research, significant findings, or novel approaches to antifungal therapies. **Exclusion Criteria:** Studies lacking robust methodologies or clear conclusions. Publications focusing solely on bacterial or viral pathogens, without relevance to fungal infections. **Screening Process:** Titles and abstracts were reviewed to ensure relevance, followed by a detailed examination of the full text. Reference lists of key articles were also screened to identify additional sources.

## GLOBAL CHALLENGES OF FUNGAL INFECTIONS

### Overview of the epidemiological landscape of global fungal infections

Fungi are widespread across a vast array of environments, including the stratosphere, arid deserts, oceanic sediments, Antarctic glaciers, and even the human gastrointestinal tract (*Naranjo-Ortiz & Gabaldon, 2019*). Although fungi account for less than 1% of the gut microbiota, recent studies emphasize their pivotal role in promoting immune responses, forming not only symbiotic but also mutualistic relationships with the host (*Arumugam et al., 2011*). Virulence determinants in human-pathogenic fungi can emerge through environmental and commensal interactions, further evolving during host colonization and infection (*Siscar-Lewin, Hube & Brunke, 2022*).

Each year, invasive aspergillosis is diagnosed in over 2.1 million individuals with conditions such as chronic obstructive pulmonary disease, cancer, or hematological malignancies, leading to around 1.8 million deaths (85.2% mortality). The annual incidence of chronic pulmonary aspergillosis reaches 1,837,272 cases, causing 340,000 deaths (18.5%). *Candida* bloodstream infections, or invasive *candidiasis*, affect approximately 1.56 million people annually, with a death toll of 995,000 (63.6%). *Pneumocystis* pneumonia strikes 505,000 individuals, resulting in 214,000 deaths (42.4%), while cryptococcal meningitis impacts 194,000 people, with 147,000 fatalities (75.8%). Other significant life-threatening fungal infections affect an additional 300,000 people, contributing to 161,000 deaths (53.7%). Fungal asthma is estimated to impact 11.5 million individuals, leading to 46,000 asthma-related deaths each year. Altogether, the global burden amounts to 6.5 million invasive fungal infections annually, with 3.8 million resulting in death, of which roughly 2.5 million (68%) are directly attributed to fungal infections (*Ikuta, Mestrovic & Naghavi, 2024*).

According to the Global Action Fund for Fungal Infections, approximately one billion individuals worldwide are affected by fungal pathogens, leading to 1.6 million deaths each year. In Germany, more than 10% of the population suffers from fungal infections, with superficial cutaneous and *onychomycoses* being the most prevalent forms (*Ruhnke et al., 2015*). In France, the annual incidence of invasive fungal infections is around 5.9 per 100,000 cases, with a 27.6% mortality rate based on hospital discharge diagnoses (*Bitar et al., 2014*). In the United States, the Centers for Disease Control and Prevention (CDC) report over 2.8 million cases of antibiotic-resistant bacterial and fungal infections annually, leading to approximately 36,000 deaths (*Centers for Disease Control and Prevention (US), 2019*).

## Types of fungal infections and primary pathogens

Fungal infections are classified based on the site of invasion into four categories: superficial mycoses, cutaneous mycoses, subcutaneous mycoses, and systemic mycoses. Systemic mycoses not only affect the skin and subcutaneous tissues but also involve internal organs and tissues, potentially leading to disseminated infections, commonly referred to as invasive fungal infections. Each type of fungal infection exhibits unique characteristics depending on the site of infection and the mode of fungal entry into the host. For instance, mucormycosis can present in six distinct forms based on the anatomical location: rhinocerebral, pulmonary, cutaneous, gastrointestinal, disseminated, and atypical presentations (*Petrikkos et al., 2012*).

Predisposing factors play a significant role in determining the clinical manifestation and prognosis of fungal infections. The prevalence of systemic mycoses caused by opportunistic fungi has increased due to the widespread use of broad-spectrum antibiotics, immunosuppressive therapies, antineoplastic agents, and advancements in organ transplantation, catheter technology, and other invasive surgical interventions—especially since the emergence of AIDS. Despite shifts in the global burden of fungal diseases, six pathogens—*Aspergillus*, *Candida*, *Cryptococcus*, *Pneumocystis jirovecii*, *Histoplasma capsulatum*, and *Mucormycetes*—persist as the leading culprits of severe infections,

collectively accounting for the majority of morbidity and mortality (*Bongomin et al., 2017*). Of these, *Candida*, *Aspergillus*, and *Cryptococcus* account for approximately 90% of fatalities related to fungal infections (*Qadri et al., 2021*).

### Pathophysiological mechanisms underlying major fungal infection pathogenesis

The primary virulence factor of *Candida albicans* is its ability to undergo filamentation, transitioning into a hyphal form that enables it to invade host tissues, cause cellular damage, and breach epithelial barriers. This process is mediated by *candidalysin*, a peptide toxin associated with the hyphal structure, representing one of the few classical virulence factors identified in human pathogenic fungi (*Allert et al., 2018*; *Moyes et al., 2016*; *Naglik, Gaffen & Hube, 2019*).

    *Aspergillus fumigatus*, a major cause of both respiratory and systemic fungal infections (*Sugui et al., 2014*), exhibits significant thermotolerance, crucial for its survival in compost environments. This thermotolerance makes it resilient to human fever and adaptable to a wide range of stressors, including osmotic and oxidative stress, desiccation, and starvation, which are typical of its soil habitat (*Paulussen et al., 2017*). A key virulence factor in *Aspergillus fumigatus* is the melanin found in its spores, which aids in concealing antigens, inhibiting phagosome maturation, and defending against oxidative stress (*Gomez & Nosanchuk, 2003*; *Schmidt et al., 2020*).

    The capsule of *Cryptococcus neoforman* s is another critical virulence factor, consisting primarily of glucuronoxylomannan (GXM), galactoxylomannan (GalXM), and smaller amounts of mannoprotein (MP), with GXM accounting for over 90% of the polysaccharide content. These virulence factors not only provide structural and enzymatic benefits for pathogen survival but also interfere with the host's immune defenses by actively modulating host-specific signal transduction pathways (*O'Meara & Alspaugh, 2012*).

    Given the wide variety of clinical presentations, incidence rates, and mortality associated with fungal infections, a thorough understanding of these pathogenic mechanisms is crucial. Effective antifungal therapies are essential to reducing the morbidity and mortality caused by these infections.

## MECHANISMS OF ANTIFUNGAL THERAPY AND RESISTANCE

### Introduction of traditional antifungal medications

Presently, various antifungal medications are available for treatment. They are categorized into four classes and can be used alone or in combination to treat a wide range of fungal diseases. Combination therapy and hybrid medications are common and effective. Some common frontline antifungals include: Polyenes, like amphotericin B (AmB), disrupt cell structure by rupturing fungal cell membranes. 5-fluorouracil (5-FC), when used in combination, inhibits nucleic acid synthesis. *Cryptococcus* species are resistant. *Micafungin* and other echinocandins inhibit fungal cell wall synthesis. *Candida* species are resistant. Fluconazole and other azoles hinder metabolism by blocking ergosterol synthesis. *Candida* species are resistant.

## Mechanism of action

Triazole antifungals, like fluconazole, block the cytochrome P450 enzyme 14$\alpha$-demethylase, halting the conversion of lanosterol to ergosterol and thereby disrupting fungal cell membranes. Ergosterol is a critical component of the fungal cell membrane, and its inhibition leads to increased cellular permeability, disrupting membrane integrity. Other antifungal agents act through various mechanisms, such as hindering lanosterol synthesis, inhibiting protein synthesis, or obstructing the production of ergosterol. Additionally, drugs like AmB create pores in the fungal cell membranes, leading to the leakage of ions and small organic molecules, which ultimately results in cell death (*Joshi et al., 2024*).

## Mechanisms of resistance in antifungal therapy

Antifungal resistance is multifactorial, arising from various factors: (1) Target overexpression; (2) Limited antifungal discovery; (3) Non-compliance with prescriptions; (4) Overuse of antifungal agents; (5) Genetic mutations; (6) Poor waste disposal practices; (7) Alterations in metabolic pathways; (8) Airborne spore dispersal; (9) Repeated use of the same fungicide; and (10) Increased vulnerability in immunocompromised individuals and cancer patients (*Aperis & Mylonakis, 2006*).

Three primary mechanisms contribute to resistance in triazoles: unregulated expression of 14$\alpha$-demethylase, alterations in the triazole binding site, and the upregulation of multidrug efflux transporters. Multiple resistance mechanisms can coexist within a single strain (*Whaley et al., 2016*). Fluconazole, a commonly used triazole derivative, is considered exceptionally safe for fungal infections and has been administered to over 16 million patients, including 300,000 individuals with AIDS. This widespread use has contributed to the significant development of drug resistance in microorganisms (*Ghannoum & Rice, 1999*). Triazoles work by inhibiting 14-demethylase, an enzyme that converts lanosterol to ergosterol, thereby impairing endogenous respiration and inhibiting yeast growth (*Spampinato & Leonardi, 2013*).

Echinocandin resistance is typically associated with mutations in the hotspot region of the *FKS* catalytic subunit of $\beta$-1,3-D-glucan synthase, as well as alterations in the lipid composition around the *FKS* gene. These are the primary contributors to echinocandin resistance (*Arastehfar et al., 2021b*).

Resistance to polyenes is less common, and its mechanisms remain unclear. These drugs primarily target ergosterol, with their action involving ergosterol sequestration and reductions in membrane ergosterol levels. Mutations in the *ERG* gene contribute to both intrinsic and acquired resistance, especially in species such as *Candida auris* and *Aspergillus terreus* (*Arastehfar et al., 2021a*).

Over recent decades, the excessive use, prolonged treatment regimens, and environmental exposure to azoles, polyenes, and echinocandins have accelerated the development of resistance. The World Health Organization predicts that antimicrobial resistance (AMR) could become the leading cause of mortality by 2050 (*Antimicrobial Resistance Collaborators, 2022*). AMR presents a major threat, highlighting the urgent need for new antifungal agents. It is estimated that around two million people die each year from fungal infections. Despite the continued use of traditional antifungal therapies, resistance
has not been adequately addressed (*Fisher et al., 2022*). Thus, it is critical for each class of antifungals to innovate and develop new medications to combat this growing resistance.

Ideally, antifungal agents should have the following characteristics: (1) Minimal or manageable toxicity and side effects, ensuring a wide therapeutic window; (2) Pharmacological properties suitable for various administration routes; (3) Fungus-specific action mechanisms; (4) Preferably fungicidal effects; and (5) Broad-spectrum efficacy against different fungal species. Promising areas for future antifungal development include natural compounds, semisynthetic and synthetic compounds, nanoparticles, peptides, and therapies that leverage novel mechanisms (*Cui et al., 2022*). Nanotechnology, in particular, offers the ability to synthesize nanoparticles with inherent antimicrobial properties (*Zuniga-Miranda et al., 2023*).

# NANOPARTICLES ARE CRUCIAL IN ANTIFUNGAL TREATMENT

## Introduction to nanoparticles

Nanoparticles are widely used in the medical field for anti-infection, biomarker detection, cancer treatment, genetic engineering, and more. Figure 1 illustrates some of the main medical applications. Nanoparticles, the cornerstone of nanotechnology, typically range in size from 10 to 100 nm. The two main approaches to nanoparticle synthesis are "Bottom-Up" methods, which begin with atoms or molecules and progress to nanostructures through chemical reactions, and "Top-Down" methods, which use physical techniques to break down bulk materials into nanoparticles (Fig. 2). They can exist as solid particles or dispersions and are capable of transporting a variety of molecules, including pharmaceuticals, aptamers, peptides, antibodies, and cationic entities (*Mohanraj & Chen, 2006*). In the rapidly evolving field of research, the application of nanoparticles in antifungal therapy has garnered significant attention. Their role can be broadly categorized into two key areas: as carriers for antifungal drugs, improving drug delivery and bioavailability, and as direct antifungal agents with intrinsic antifungal properties.

## Nanoparticles as carriers for antifungal agents

The poor solubility of antifungal medications has long posed a challenge to their clinical efficacy. Several techniques, including solid lipid nanoparticles (SLNs), nanostructured lipid carriers (NLCs), liposomes, cubosomes, and herbosomes, have been developed to enhance drug solubility and bioavailability, thereby improving therapeutic outcomes. Key fungal targets include dihydrofolate reductase (DHFR), acetohydroxy-acid synthase (AHAS), farnesyltransferase, and endoglucanase. Network pharmacology has identified that voriconazole (VCZ) interacts with essential genes involved in ergosterol biosynthesis, such as lanosterol 14 $\alpha$-demethylase inhibitors (ERG11), ergosterol biosynthesis protein 5 (ERG5), and several others (*Tiwari et al., 2023*).

Nanocarrier systems are especially effective for antifungal medications that suffer from poor solubility, delivery, and absorption (*Tiwari et al., 2022*), such as VCZ. NLCs have been shown to improve VCZ's solubility and stability. Research on ocular fungal infections demonstrates that an antifungal *in situ* gel with VCZ-loaded NLCs significantly enhances
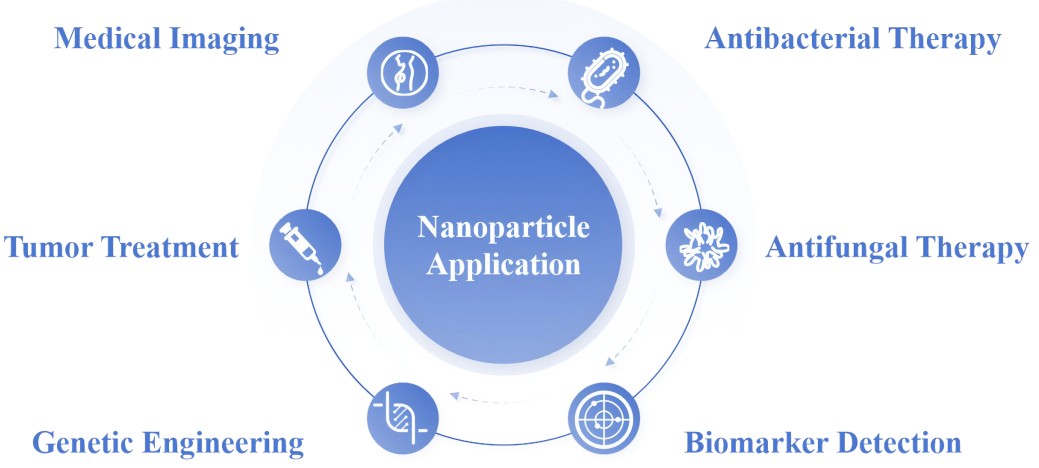

**Figure 1  Diverse applications of nanoparticles in various fields.** Nanoparticle application is surrounded by six key application areas.

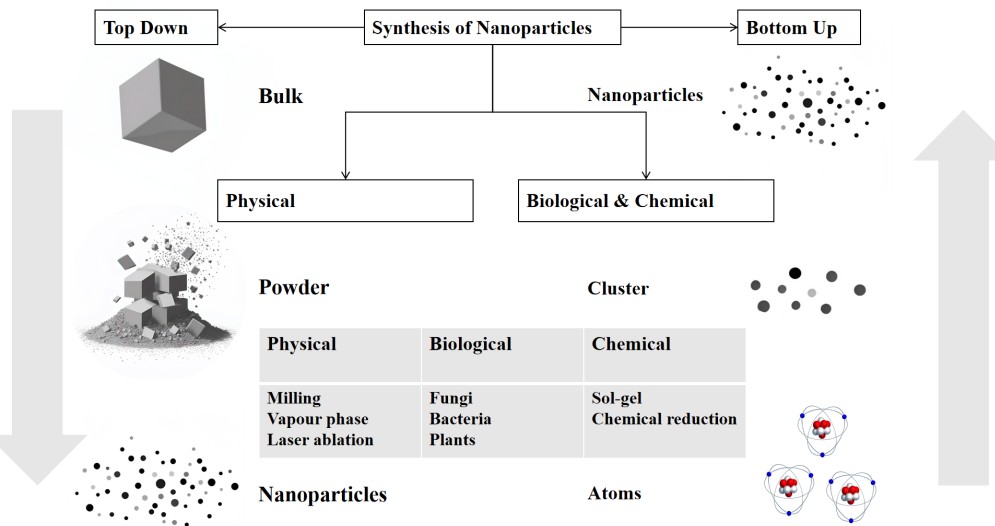

**Figure 2  Schematic representation of the top–down and bottom–up synthesis methods of nanoparticles.** There are two main approaches: the top–down method, which starts with bulk materials and breaks them down into smaller particles such as powder and finally nanoparticles through physical processes like milling, vapour–phase, and laser ablation; and the bottom–up method, which constructs nanoparticles from atoms or clusters using biological (fungi, bacteria, plants) and chemical (sol–gel, chemical reduction) processes. The diagram illustrates the different pathways and the intermediate states in the synthesis of nanoparticles.

solubility and bioavailability, while effectively prolonging drug release, offering a novel therapeutic approach (*Tiwari et al., 2023*). Similarly, studies on itraconazole (ITZ)-loaded polymeric nanoparticles (ITZ-NPs) have shown enhanced sustained drug release and superior antifungal efficacy. These nanoparticles exhibit minimal hemolysis and negligible

**Table 1 Comparison of traditional *vs* nanoparticle-based antifungal therapies.**

| Criteria | Traditional therapies | Nanoparticle-based therapies |
|---|---|---|
| Solubility | Low | Improved with carriers (*e.g.*, liposomes) |
| Bioavailability | Poor | Enhanced |
| Toxicity | Higher toxicity | Reduced with controlled release |
| Targeted delivery | Non-specific | Targeted, site-specific delivery |
| Resistance mechanisms | Limited by genetic mutations and efflux pumps | More effective at overcoming resistance |

venous irritation, indicating greater biocompatibility compared to commercially available cyclodextrin-based formulations (*Ling et al., 2016*).

AmB, a potent antifungal agent, presents several clinical challenges, including low solubility, poor bioavailability, sensitivity to acidic environments, limited gastrointestinal absorption, and varying toxicity depending on its aggregation state. It also requires parenteral delivery and careful storage (*Sharifi et al., 2024*). These complexities limit its oral efficacy, as AmB demonstrates poor intestinal permeability and absorption at physiological pH. Recently, SLNs have emerged as a promising platform for oral AmB delivery, with temperature-stable lipid cores stabilized by surfactants or emulsifiers. These nanoparticles minimize renal distribution of AmB, reduce toxicity, and modulate drug release effectively (*Fairuz, Nair & Billa, 2022*). SLNs can maintain AmB solubility in gastrointestinal fluids by inducing the secretion of bile salts and phospholipids, forming mixed micelles to enhance lymphatic absorption in the small intestine (*Zaioncz, Khalil & Mainardes, 2017*). Additionally, gliadin/casein nanoparticles(AmB-GliCas NPs) have been developed to mitigate cytotoxicity of AmB. These nanoparticles provide sustained release over 96 h and exhibit stability in simulated gastrointestinal fluids, significantly reducing hemolysis and cytotoxicity compared to free AmB (*Marcano et al., 2024*). AmB nanoparticles have also demonstrated increased efficacy against leishmaniasis (*Firouzeh, Asadi & Tavakoli Kareshk, 2021*).

In a study investigating the delivery of itraconazole to skin lesions *via* NLCs, oral administration led to a 23–36% reduction in transepidermal water loss and a twofold reduction in transdermal delivery, highlighting the effectiveness of targeted delivery to the skin without compromising antifungal efficacy (*Passos et al., 2020*).

Further expanding the possibilities of nanoparticle-mediated delivery, cell-mediated nanoparticle delivery systems (CMNDDs) leverage the natural properties of various cell types, such as the homing capabilities of stem cells, the chemotaxis of neutrophils, the prolonged circulation of erythrocytes, and the internalization capacity of macrophages. CMNDDs can extend nanoparticle circulation, traverse biological barriers such as the blood–brain and bone marrow–blood barriers, and rapidly reach target areas, making them highly promising for precision antifungal therapy (*Cheng & Wang, 2024*). Table 1 summarizes the comparison between traditional and nanoparticle-based antifungal therapies.

## Nanoparticles as direct antifungal agents

The cytotoxic effects of nanomaterials, particularly concerning ROS, are primarily due to the disruption of the respiratory chain or direct induction by the nanomaterials. A surge in ROS, caused by acute oxidative stress, can damage various cell types, inhibiting lipid peroxidation, altering proteins and enzymes, and causing RNA and DNA damage. While high concentrations of ROS can lead to cell death, lower concentrations may cause severe DNA damage and mutations (*Gupta, 2021*).

Biogenic silver nanoparticles (AgNPs) inhibit $\beta$-glucan synthesis, compromising the structural integrity of *Candida albicans* cell wall and reducing its resilience. AgNPs generate ROS, leading to mitochondrial dysfunction, apoptosis, DNA fragmentation, and activation of metacaspases. They also modulate the *Ras*-mediated signaling cascade in *Candida albicans* by downregulating key genes such as *ECE1*, *TEC1*, *TUP1*, and *RFG1*, which are essential for the yeast-to-hyphae transition (*Halbandge et al., 2019*). AgNPs further exert fungistatic effects by disrupting $\beta$-glucan synthase, compromising cell wall architecture and diminishing mechanical resistance. ROS produced by AgNPs induce mitochondrial dysfunction, triggering apoptosis, phosphatidylserine externalization, nuclear DNA fragmentation, and metacaspase activation (*Mba & Nweze, 2020*).

The antifungal mechanisms of metal nanoparticles extend beyond AgNPs and can be categorized as follows. The antifungal mechanisms of nanoparticles are depicted in Fig. 3. These mechanisms mainly involve the generation of ROS that can damage fungal cell walls and membranes, leading to cell lysis and the leakage of intracellular contents

(1) Size effect: nanoparticles can physically disrupt fungal cell walls and membranes, causing leakage of cellular contents and cell death. For example, gold nanoclusters (Au NCs) smaller than 2 nm exhibit unique antimicrobial properties. Research has shown that shrinking Au nanoparticles below this size imparts them with broad-spectrum antimicrobial activity, killing both Gram-positive and Gram-negative bacteria by increasing ROS production and disrupting bacterial metabolism (*Zheng et al., 2017*).

(2) Surface characteristics: nanoparticles with rough or sharp surfaces are more effective at penetrating and rupturing fungal cell walls and membranes. *Wang et al. (2018)* developed mesoporous silica nanospheres with rough surfaces that enhanced adhesion to bacterial surfaces through multivalent interactions. Similarly, spiky $TiO_2$ particles can penetrate cell membranes, allowing for the direct release of biomolecules into the cytosol (*Wang et al., 2017*).

(3) Charge effect: positively or negatively charged nanoparticles interact with the charged components of fungal cell walls and membranes, causing membrane disruption *via* electrostatic interactions. Nanostructures with dense positive charges have demonstrated more efficient cellular uptake, significantly enhancing their ability to penetrate three-dimensional multicellular spheroids (*Shi et al., 2022*).

(4) Shape effect: the shape of metal nanoparticles, such as spherical, rod-like, or star-shaped, influences their interaction with cell walls. *Elbourne et al. (2020)* developed gallium-based liquid metal nanoparticles (GLM-Fe) using an iron-gallium-indium-tin alloy. Under a magnetic field, these nanoparticles change shape and penetrate bacterial

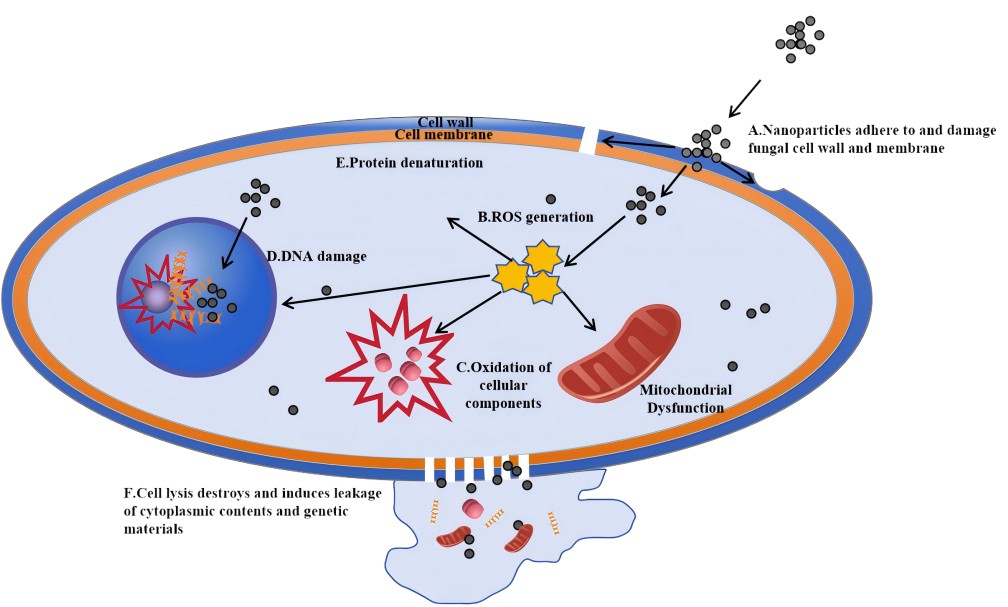

**Figure 3** **Antifungal mechanisms of nanoparticles: a cellular-level perspective.** Nanoparticles first interact with the cell wall and cell membrane of fungal cells. (A) Reactive oxygen species (ROS) generated by nanoparticles damage the cell wall, and the nanoparticles adhere to and dissolve the membrane, facilitating their entry into the cell. (B) Once inside, nanoparticles can trigger the generation of more ROS. (C) These ROS then cause oxidation of cellular components, such as those in mitochondria. (D) Nanoparticles can also directly damage the DNA within the cell nucleus. Eventually, (E) cell lysis occurs, destroying the cell and inducing the leakage of cytoplasmic contents and genetic materials, which ultimately leads to the death of the fungal cell.

biofilms, killing over 95% of *Pseudomonas aeruginosa* and *Staphylococcus aureus* within 90 min (*Elbourne et al., 2020*).

(5) Aggregation phenomenon: metal nanoparticles can aggregate on cell surfaces, increasing physical pressure on cell walls and membranes. Janus nanoparticles (asymmetric particles with dual surface properties) with optimized protrusion sizes have been shown to rapidly anchor to and penetrate cell membranes, improving uptake efficiency (*Xia et al., 2024*). *Linklater et al. (2020)* demonstrated that increased membrane tension caused by nanoparticle interaction is a universal phenomenon across bacterial strains.

(6) Mechanical stress: nanoparticles can exert mechanical stress on fungal cell walls and membranes, leading to deformation or rupture. *Liu et al. (2023)* developed a biophysical model showing that nanoparticle adsorption induces stretching and compression of membranes, causing mechanical stress.

(7) Cell wall remodeling: metal nanoparticles may disrupt fungal cell wall biosynthesis, leading to structural defects and instability.

(8) Alterations in membrane fluidity: *Paul, Pandey & Neogi (2023)* demonstrated how NiO and CuO-NiO mixed metal oxide nanoparticles alter bacterial cell membranes, increasing permeability and inducing oxidative stress, which destabilizes the membrane.

(9)   Cellular endocytosis: metal nanoparticles can be internalized by fungal cells through endocytosis, increasing intracellular pressure and causing organelle damage.

(10)  Biological membrane penetration: studies on AgNPs using staining and imaging techniques have shown that AgNPs can disrupt fungal hyphal structures and cause organelle degradation, further confirming their toxic effects on fungal cells (*Jian et al., 2022*).

## Advantages of nanoparticle-based drug delivery

Nanoparticle-based drug delivery offers several key advantages: (1) improved delivery of poorly water-soluble drugs; (2) targeted drug delivery to specific cells or tissues; (3) transcytosis across tight epithelial and endothelial barriers; (4) delivery of large macromolecular drugs to intracellular targets; (5) simultaneous delivery of multiple drugs or therapeutic modalities for combination therapies; (6) visualization of drug delivery sites *via* imaging modalities; and (7) real-time assessment of a therapeutic agent's *in vivo* efficacy (*Farokhzad & Langer, 2009*).

One innovative approach to enhancing the bioactivity of small-molecule antifungal agents involves metal nanozyme engineering. For example, AmB-conjugated gold nanoparticles (AmB@AuNPs), synthesized through a simple one-step process, exhibit remarkable peroxidase-like enzymatic activity in $H_2O_2$-mediated reactions. Their maximal catalytic rate (V_max) is 3.4 times greater than that of unmodified AuNPs. This enzyme-mimetic activity significantly enhances the fungicidal efficacy of AmB. Consequently, the minimum inhibitory concentrations (MICs) of AmB@AuNPs against *Candida albicans* and *Saccharomyces cerevisiae W303* are reduced by 1.6-fold and 50-fold, respectively, compared to AmB alone (*Jiang et al., 2024*).

Nanoparticle-based drugs also present a novel strategy for tackling drug-resistant fungi. Antibiotics typically act through mechanisms such as enzyme inhibition, membrane structure modification, and interference with transcription and translation (*Wahab et al., 2021*). *Lotfali et al. (2021)* demonstrated that nanoparticles can alter the cell wall structure of resistant strains by forming pores, potentially increasing the susceptibility of resistant strains. They synthesized Ag-NPs, which exhibited superior antifungal efficacy against resistant *C. glabrata* strains when compared to selenium (Se-NPs) and AuNPs.

In another example, *Tran et al. (2017)* employed a simple and eco-friendly method to synthesize biocompatible hybrids containing copper oxide nanoparticles (CuONPs) from cellulose (CEL) and chitosan (CS) or CEL and keratin (KER). These composites showed significant antibacterial activity against a broad range of bacteria and fungi. Importantly, the antimicrobial performance of the composite was positively correlated with CuONP concentration, and it remained compatible with human fibroblast cells at concentrations of 35 nmol/mg or lower (*Tran et al., 2017*). Additionally, pomegranate peel aqueous extract facilitated the synthesis of highly antibacterial iron oxide nanoparticles, showcasing the antimicrobial potential of plant-derived nanoparticles (*Sharma et al., 2022*).

Furthermore, a study evaluating curcumin-coated silver nanoparticles (Cur-Ag NPs) synthesized through environmentally friendly methods demonstrated significant antifungal efficacy against a panel of *Candida* and *Aspergillus* species. Cur-Ag NPs exhibited strong
**Table 2  Comparison of advantages and disadvantages of various types of nanoparticles.**

| Nanoparticle type | Advantages | Disadvantages | Reference |
|---|---|---|---|
| Silica nanoparticles | Excellent chemical and thermal stability, good biocompatibility | The agglomeration of nanoparticles affect antibacterial activity | *Singh et al. (2017)* |
| Metal and metal oxide nanoparticles | High-efficiency antibacterial activity, multi-target mechanism of action | The limitations of size and morphology affect antibacterial activity | *Massoudi et al. (2022)*, *Aderibigbe (2017)* |
| Polymeric nanoparticles | The ability to enhance drug stability, the ability of adjustability and targeted delivery | Toxic degradation, toxic monomers aggregation, residual material | *Sivadasan et al. (2021)*, *Lakshminarayanan et al. (2018)* |
| Lipid nanoparticles | The ability to enhance drug solubility and bioavailability, significant advantages of local antifungal treatment and skin permeability | Polymorphic transformation may occur during storage, leading to drug payload loss | *Singulani et al. (2018)*, *Ramu, Spandana & Preethi (2021)* |
| Smart nanoparticles | Precision medicine and prolonged drug release | The complex manufacturing process and the high costs | *Meng et al. (2023)* |
| Phage nanoparticles | Low environmental impact, fewer side-effects, narrow antimicrobial spectrum, effective agains biofilms | Adaptive anti-phage immunity may develop through multiple dosing, lack of standardized guidelines for dosage and administration, limited experience in the application for fungal infections. | *Ling et al. (2022)*, *Zhang et al. (2022b)*, *Manohar et al. (2024)* |

antifungal properties, particularly against azole-resistant strains of *Aspergillus* and *Candida*, underscoring their potential to combat drug-resistant fungal infections (*Amini et al., 2023*).

## Advances in nanoparticle-based antifungal therapy

Significant progress has been made in various research directions of nanoparticle antifungal therapy, demonstrating reliable efficacy in both local and systemic infections. Recent studies have demonstrated that the AgCu2O nanoparticles (AgCuE NPs) gel, a nanoparticle formulation, showed good biosafety and no obvious ophthalmic and systemic side effects (*Ye et al., 2022*). High efficacy in treating deep cutaneous and onychomycosis fungal infection has also been demonstrated (*Wang et al., 2023a*). Nanoparticles have emerged as effective drug carriers for treating invasive fungal infections in immunocompromised patients, significantly enhancing drug bioavailability and reducing side effects (*Botero Aguirre & Restrepo Hamid, 2015*). Comparison of advantages and disadvantages of various types of nanoparticles was presented in Table 2. Various types of nanoparticles and their newly applications were summarized below.

(1) **Silica Nanoparticles:** Recent progress has been made in nanoparticle drug co-delivery and precision targeting. pH-sensitive gated mesoporous silica nanoparticles enhanced antifungal efficacy of tebuconazole and amine-functionalized silica nanoparticles enhanced topical econazole antifungal efficacy (*Mas et al., 2014*). In 2022, a study optimized mesoporous silica particles (MSNs) co-loaded with ketoconazole and betamethasone (EN-TA-MSNs) using a central composite rotatable design (CCRD), which significantly enhancing drug release efficiency (ketoconazole 68%, betamethasone 70%). This system also reduced skin irritation (erythema grade was reduced fourfold) and promoted wound

healing. Animal experiments demonstrated superior *in vivo* antifungal efficacy compared to pure drugs, with cell viability increased to 90% (*Maheen et al., 2022*).

**(2) Metal and Metal Oxide Nanoparticles:** The broad spectrum of target fungi for metal nanoparticles has suggested a significant demand and market potential. AgNPs exerted broad-spectrum antifungal effects against *Candida*, *Aspergillus,* and other fungi by disrupting fungal cell membranes by releasing silver ions or ROS (*Halbandge et al., 2019*). Iron oxide nanoparticles (FeNPs) suppress a number of fungal infections, including *Fusarium solani*, *Aspergillus niger*, and *Candida albicans* (*Nehra et al., 2018*). Recent advances in the research of metal nanoparticles synthesized by fungi have demonstrated their advantages of high yield and environmental friendliness, making them one of the hotspots in current research (*Cruz et al., 2024*).

**(3) Polymeric Nanoparticles:** Multiple studies have developed Chitosan-PNs for disrupting biofilms and loading anti-drug agents. Chitosan-PNs significantly inhibited *Candida* biofilm formation and decreased the number of colony forming unit (CFU) of *Candida spp* (*Gondim et al., 2018*). Chitosan-PNs loaded with P10 peptide significantly reduced fungal burden in the lungs of mice *via* intranasal delivery (51 days post-infection), with an effective dose of 1 µg to decrease CFU (*Rodrigues Dos Santos Junior et al., 2020*).

**(4) Lipid Nanoparticles:** Lipid-based nanoparticles and lipid-structured material nanoparticles have shown significant potential in the treatment of skin fungal infections. The *in vitro* results showed that NEA exhibited better antifungal activity than free AmB in both planktonic and sessile cells, with >31% inhibition of mature biofilm (*Marena et al., 2023*). Moreover, compared to traditional SLNs, NLCs significantly reduce drug leakage through the design of a disordered crystalline matrix, achieving an encapsulation efficiency of over 80% and enhancing the controllability of drug release (*Dudhipala & Ay, 2020*).

**(5) Smart Nanoparticles:** With the development of precision medicine and the increasing demand for controllable drug release, the research on intelligent and smart nanoparticles has been put on the agenda. Responsive nanoparticles (such as pH-sensitive or enzyme-sensitive types) can precisely release drugs in the infected microenvironment, thereby intelligently controlling and slowing the release of antimicrobial agents (*Meng et al., 2023*).

**(6) Phage Nanoparticles**: Phages displaying specific peptides can conjugate with nanoparticles, combining the benefits of peptides and nanomaterials for precise fungal detection. Additionally, phage nanomaterials as carriers can reduce drug toxicity, prolong drug circulation, stimulate immune responses, and possess inherent antifungal effects (*Xu et al., 2022*).

## Challenges and difficulties in nanoparticle-based antifungal therapy

Nanoparticles have shown great potential in the treatment of fungal infections, but still face multiple difficulties and challenges, mainly involving the following aspects.

(1) **Safety and Toxicity Issues:** Concerns about the toxicity of nanomaterials were raised many years ago, as reported, inhalation of nanoparticles can trigger inflammation and fibrosis, especially carbon nanotubes, which may deposit and cause long-term lung damage (*Oberdorster, Oberdorster & Oberdorster, 2005*). In recent years, research has focused on the reproductive toxicity of nanoparticles. *In vivo* and *in vitro* studies show that polylactic

acid microplastic (PLA-MPs) derived nanoparticles can penetrate the blood-testis barrier (BTB) and localize in the spermatogenic microenvironment. Long-term exposure to PLA-MPs causes significant reproductive toxicity in mice, characterized by reduced sperm concentration and motility, increased sperm deformity rates, and disrupted sex hormone levels (*Zhao et al., 2025*). The toxicity of nanomaterials in the reproductive system may affect the placental barrier, leading to fetal abnormalities (*Ahmad, 2022*). To address this issue, further mining and analysis of existing literature data are extremely important. For instance, *Huang et al. (2025)* used machine learning and SHAP analysis to identify *IL-1β* in THP-1 cells as an *in vitro* biomarker for nanoparticle-induced pulmonary toxicity . Futhermore, the cytotoxicity of nanomaterials can be significantly reduced through surface chemical modification, a method that has been validated in multiple studies (*Attarilar et al., 2020*; *Rivas et al., 2022*).

**(2) Scalability of Synthesis:** The most challenging steps in the development of nanomedicine products come from the transition from laboratory-scale batches to large-scale industrial batches, as well as the selection of excipients required for the production of high-quality drugs. Traditional methods such as phase separation were only suitable for small-scale production, relying on organic solvents, and were limited in particle size (50–500 nm). Milling is cost-effective and suitable for large-scale production, but it has issues such as poor control over the shape of nanoparticles and the need for cooling due to heat generation (*DeFrates et al., 2018*). Microbial synthesis of nanoparticles faces challenges such as polydispersity, low yield, and aggregation, and requires extreme conditions (*e.g.*, specific temperature, pH) that increase process complexity (*Khan et al., 2023*). Recent studies have shown optimizations in batch production and quality control of nanomaterials. Breakthroughs in microfluidic technology process optimization and enables the efficient and uniform preparation of nanoparticles through precise control of mixing conditions. For example, the microfluidic synthesis of lipid nanoparticles (LNPs) for COVID-19 mRNA vaccines supports high-throughput production at a rate of 25 kg/h (*Li et al., 2023*). Continuous nanoprecipitation of SLNs *via* static mixers, achieves a production rate of 150 g per hour (*Gautam, Kim & Yong, 2021*). Achieving a batch-to-batch variation control of 16% in the production of HIV antigen peptide chitosan nanoparticles using process analytical technology (PAT) and quality by design (QbD) (*Klein et al., 2020*).

**(3) Shortcomings in Clinical Translation and Pharmacodynamics Research:** Most current studies remain at the *in vitro* or animal experimental stage, lacking systematic *in vivo* pharmacokinetic data (such as biodistribution and metabolic pathways (*Dordevic et al., 2022*; *Metselaar & Lammers, 2020*). The dynamic relationship between the drug release behavior of nanoparticles (such as extended release and burst release) and antifungal activity lacks quantitative research. For example, although the release of miconazole nitrate polymer nanoparticles conforms to the Korsmeyer-Peppas release model, the correlation between its *in vivo* antifungal efficacy and release rate still needs to be verified (*Bresinskya & Goepfericha, 2025*). Establishing a database of "structure–activity–toxicity" for nanoparticles and using artificial intelligence to predict their biodistribution and efficacy may be helpful to solve this problem.

**(4) Biofilm Resistance and Biocompatibility Challenges:** The three-dimensional structure of fungal biofilms is complex, and the extracellular polysaccharide matrix they secrete can hinder drug penetration, significantly reducing therapeutic efficacy (*Kowalski et al., 2020*). Although AgNPs can disrupt the cell membrane and induce ROS accumulation to kill fungi, more research needs to focus on nanomaterials designed to cross different physiological barriers, effectively addressing challenges posed by skin, corneal, and blood–brain barriers (*Liu et al., 2024*). As a typical example, Wang's team invented a $CuFeSe_2$-PVP nano-blade, which has achieved encouraging *in vivo* antifungal therapeutic effects and exhibited excellent biocompatibility (*Wang, Zhou & Wang, 2023b*).

**(5) Resistance challenges:** Strikingly, a study showed that rather than eradicating persister cells, a wide range of nanoparticles promote the formation of bacterial persistence (*Zhang et al., 2022a*). Although this mechanism primarily targets bacteria, fungi may also exhibit similar phenotypic heterogeneity, which requires further investigation. The design of nanomaterials needs to be dynamically optimized to address these challenges (*Liu et al., 2024*).

## CONCLUSION AND FUTURE PERSPECTIVES

This review comprehensively explores the current status and research progress of nanoparticle-based therapies for fungal infections. Fungal infections pose a serious threat to human health, resulting in a significant number of fatalities each year. The growing issue of antifungal resistance further complicates treatment, highlighting the need for novel therapeutic strategies. Nanoparticles, with their unique properties, have demonstrated great potential as both carriers for antifungal drugs and as direct antifungal agents. The innovative application of nanoparticles to enhance the efficacy of traditional antifungals and combat drug-resistant strains is a particularly promising research avenue.

First, the diversity of fungal pathogens and the complexity of their interactions with the host demand a multifaceted approach to treatment. Nanoparticles, owing to their high surface area-to-volume ratio, tunable size, and modifiable surfaces, provide a versatile platform for drug delivery systems. They can improve the solubility and bioavailability of poorly water-soluble antifungal drugs, enable targeted delivery to infection sites, and facilitate the crossing of biological barriers.

Second, nanoparticles exhibit diverse mechanisms of action as antifungal agents. These include the generation of ROS disruption of cell walls, and interference with fungal cell membranes integrity. Physical properties such as nanoparticle size, shape, and surface characteristics play a crucial role in determining their antifungal activity.

However, several challenges remain. The potential toxicity of nanoparticles to mammalian cells, the need for controlled and sustained drug release systems, and the scalability of nanoparticle synthesis for clinical applications all require further investigation. Additionally, the interactions between nanoparticles and the immune system, as well as their long-term effects in the body, are not yet fully understood. Major challenges and solutions in nanoparticle-based therapies are summarized in Table 3.

For future perspectives, as Artificial Intelligence (AI) technologies have already gained considerable research and development in the medical field. In the face of the challenges

**Table 3  Challenges and solutions in nanoparticle-based therapies.**

| Challenges | Research solutions |
| --- | --- |
| Potential toxicity to mammalian cells | Surface modifications to reduce toxicity |
| Scalability of synthesis | Eco-friendly, large-scale synthesis methods |
| Controlled and sustained drug release | Development of new nanocarrier systems |
| Biocompatibility | Biocompatible coatings |
| Overcoming multi-drug resistance | Synergistic approaches with existing antifungals |

mentioned above in nanotechnology, AI algorithms can analyze the physicochemical properties of nanoparticles (such as size, shape, and surface modification) and predict their potential cytotoxicity and immunogenicity. For instance, metal oxide nanoparticle (MONP) biocompatibility has been accurately predicted by AI models, serving as a benchmark for evaluating other nanoparticles (*Soltani et al., 2021*). Specifically, AI can optimize drug and dose parameters in combinatorial nanomedicine to fully realize its potential (*Ho, Wang & Kee, 2019*). These developments may improve the safety and clinical translation of nanomedicines, hastening the development of nanotechnology.

Looking forward, it is imperative to continue investigating the full potential and limitations of nanoparticle-based antifungal therapies. Future research should focus on the following key areas:

(1) Elucidating the mechanisms behind the antifungal efficacy of various nanoparticles to develop more effective therapeutic agents.

(2) Assessing the safety and biocompatibility profiles of nanoparticles in both preclinical and clinical settings.

(3) Developing environmentally sustainable methods for nanoparticle synthesis.

(4) Establishing methodologies for large-scale synthesis of nanoparticles with uniform quality and characteristics.

(5) Exploring synergistic approaches that combine nanoparticles with existing antifungal drugs to address resistance and optimize therapeutic outcomes.

(6) Integrating AI technologies provides more breakthroughs and enhances the efficiency of nanomedicine development.

The incorporation of nanotechnology into antifungal treatments represents a promising frontier in the fight against fungal infections. With sustained research and technological advances, nanoparticle-based interventions are expected to play an increasingly pivotal role in the clinical management of these challenging diseases.

## ACKNOWLEDGEMENTS

We sincerely acknowledge the contributions of the OpenAI team for developing ChatGPT, which we used as a language enhancement tool during the preparation of this manuscript. ChatGPT assisted us in refining the tone, correcting singular/plural forms, identifying incorrect word usage, and improving overall coherence, significantly enhancing the linguistic quality of our work.

### Funding

This work was supported by Shenzhen High-level Hospital Construction Fund (No. G2022120). The funders had no role in study design, data collection and analysis, decision to publish, or preparation of the manuscript.

### Grant Disclosures

The following grant information was disclosed by the authors:
Shenzhen High-level Hospital Construction Fund: G2022120.

### Competing Interests

The authors declare there are no competing interests.

### Author Contributions

- Qinglin Wu conceived and designed the experiments, performed the experiments, analyzed the data, prepared figures and/or tables, and approved the final draft.
- Fulan Cen conceived and designed the experiments, performed the experiments, analyzed the data, prepared figures and/or tables, and approved the final draft.
- Ying Xie analyzed the data, prepared figures and/or tables, and approved the final draft.
- Xianjia Ning analyzed the data, authored or reviewed drafts of the article, and approved the final draft.
- Jinghua Wang analyzed the data, authored or reviewed drafts of the article, and approved the final draft.
- Zhenghao Lin conceived and designed the experiments, performed the experiments, analyzed the data, prepared figures and/or tables, authored or reviewed drafts of the article, and approved the final draft.
- Jia Huang conceived and designed the experiments, performed the experiments, analyzed the data, prepared figures and/or tables, authored or reviewed drafts of the article, and approved the final draft.

### Data Availability

This is a literature review.

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
