# Peer review of "Nanoparticle-based antifungal therapies innovations mechanisms and future prospects"

_PeerJ, doi:10.7717/peerj.19199_

## Round 0.1 · original submission · Major Revisions

The reviewers offered helpful suggestions for improving the manuscript. The reviewers recommended minor edits for readability, expanding discussions on nanoparticles and their challenges, and suggested adding nanoparticle synthesis and antifungal mechanism diagrams, including SEM images and a flow chart for medical applications. They also advised clarifying the review’s purpose, adding citations, italicizing scientific names, removing redundant keywords, restructuring the introduction and methodology, and ensuring grammatical accuracy.

·

Basic reporting

The overall grammar and structure of the manuscript are well-written. However, minor edits in some sections, particularly lines 111–115 and 156–158, could enhance readability, clarity, and flow.

The introduction is comprehensive and provides substantial information on fungi, types of infections, and antifungal therapies. However, the section addressing nanoparticles and recent advances appears overly summarized. This brevity detracts from the purpose of the review, especially if its primary focus is intended to be on recent advancements, challenges, and future directions.

For example, Sections 3.3.2 and 3.3 could benefit from more detailed discussions of the progress made in using nanoparticles for antifungal therapies. It would be helpful to elaborate on specific studies, the results achieved, and comparisons between different approaches or nanoparticles. Additionally, highlighting how certain challenges faced by earlier nanoparticle systems were addressed or overcome would strengthen the discussion.

The section on challenges appears underdeveloped, with only one paragraph addressing this critical aspect (lines 373–378). Given that one of the stated objectives of the review is to detail the challenges faced, this section would benefit from greater depth. It would be helpful to elaborate on specific challenges associated with some of the innovations in nanoparticles, providing detailed insights supported by examples from the reviewed literature.

Experimental design

No comment

Validity of the findings

No comment

·

Basic reporting

Thank you for your email and opportunity providing me to review manuscript. The manuscript looks good but still need further improvement. Here please find my detailed comments and suggestions.
Major comments
1) This review paper lack method of synthesis of nanoparticles with diagram.
2) I suggest authors to add SEM images that show the effects of nanoparticles including control and treatment.
3) why this review paper is written?
4) authors need to add diagram showing antifungal mechanism.
5) Authors need to add a flow chart showing application of nanoparticles regarding medical.
Minor corrections
Line 20: please mention annually mortality rate.
Line 34: Please remove the key words that are already used in title. I suggest you to choose unique key words.
Line 38-46: I can not see any citation in the text. I suggest authors to add relevant citations here.
There are two headings of introduction. I suggest you to mention only one introduction. Methodology part need to move last before conclusion.


There are many scientific name are not italicized. Please go through carefully and do corrections.

Experimental design

Looks fine

Validity of the findings

Need to improve

Additional comments

English editing and grammatical corrections needed.

---

## Round 0.2 · accepted · Accept

The authors adequately addressed the reviewers' comments.

·

Basic reporting

I appreciate the authors’ efforts in addressing my earlier comments. The manuscript is well-structured and provides a comprehensive introduction, a thorough discussion of recent advances in nanoparticle-based therapies targeting fungal infections, and a critical analysis of the associated challenges. Additionally, the discussion of future directions adds valuable insights to the field.

The figures are well-designed and effectively complement the text, while the overall language and presentation are clear and appropriate for publication. Given these strengths and the improvements made, I recommend the manuscript for publication.

Experimental design

No comment

Validity of the findings

No comment

·

Basic reporting

I accept this paper in current form but I recommend authors to check whole manuscript grammatically.
Thank you

Experimental design

Matched

Validity of the findings

Accurate

Additional comments

I accept this paper